# Positive-Strand RNA Viruses Induce LTR Retrotransposon Transcription and Extrachromosomal Circular DNA Generation in Plants

**DOI:** 10.3390/ijms27010286

**Published:** 2025-12-26

**Authors:** Pavel Merkulov, Anna Bolotina, Anastasia Vlasova, Anna Ivakhnenko, Alena Prokofeva, Danil Perevozchikov, Elizaveta Kamarauli, Alexander Soloviev, Ilya Kirov

**Affiliations:** 1Moscow Center for Advanced Studies, Kulakova Str. 20, 123592 Moscow, Russia; paulmerkulov97@gmail.com (P.M.); boloti.anya@yandex.ru (A.B.); vlasova.nactia@yandex.ru (A.V.); anna22072015@gmail.com (A.I.); prokofeva.ai@phystech.edu (A.P.); danilperez@mail.ru (D.P.); kamarauli2001@yandex.ru (E.K.); 2All-Russia Research Institute of Agricultural Biotechnology, Timiryazevskaya Str. 42, 127550 Moscow, Russia; a.soloviev70@gmail.com; 3All-Russia Center for Plant Quarantine, Pogranichnaya Str. 32, 140150 Ramenskoe, Russia

**Keywords:** LTR-retrotransposons, plant RNA viruses, mobilome, transposition, extrachromosomal circular DNA, Tobacco rattle virus, Potato virus X, Tobacco ringspot virus, Oxford nanopore sequencing

## Abstract

Mobile elements, particularly long terminal repeat retrotransposons (LTR-RTEs), are abundant and dynamic components of plant genomes. Although viral infections are known to transcriptionally activate retrotransposons, it remains unclear whether such virus-induced activation leads to their mobilization. To address this question, we examined LTR-RTE activation in *Arabidopsis thaliana*, *Brassica napus*, and *Nicotiana benthamiana* following infection with the RNA viruses Tobacco rattle virus (TRV), Potato virus X (PVX), and Tobacco ringspot virus (TRSV). Nanopore cDNA sequencing revealed virus-specific transcriptional responses, with PVX uniquely triggering a strong transcriptional burst of diverse LTR-RTE families in *N. benthamiana*. To test the role of viral suppressors of RNA silencing (VSRs) in this process, we analyzed extrachromosomal circular DNA (eccDNA) from plants infected with TRV expressing the VSR P19. This analysis identified eccDNA derived from *Ty3*/*Gypsy Galadriel* elements, demonstrating that viral infection can promote not only retrotransposon transcription but also eccDNA production, which may indicate the ability of LTR-RTEs to transpose. These findings clearly illustrate that plant–virus interactions can induce not only changes in gene transcription, but also the activation of multiple retrotransposons, highlighting a potential evolutionary interface linking antiviral defense and transposon regulation.

## 1. Introduction

Plants are sessile organisms that constantly face various abiotic and biotic stresses. Unable to physically evade these stresses, they must adapt through the gradual process of natural selection acting on the genetic diversity of their progeny. The relationship between biotic stresses and genome changes remains an open question. It is intriguing to understand the evolutionarily shaped molecular mechanisms that stimulate genomic alterations in response to stress, thereby enhancing plant genetic diversity for subsequent natural selection. Although mutation distribution across the genome is generally random, both mutation and recombination rates can be modulated by stress conditions. Several studies have shown that bacterial- and virus-induced stresses can trigger genome rearrangements via increased production of reactive radicals [1,2,3]. Furthermore, stress-induced genetic changes can be inherited by subsequent generations, as demonstrated in *Arabidopsis* and tobacco [3,4]. These findings highlight the existence of molecular interfaces linking biotic stress responses and genomic alterations, which may provide additional sources of genetic diversity. One such interface involves stress-induced changes in DNA methylation [5]. Wilkinson et al. (2019) [5], using R-genes as examples, proposed that hypermethylation at specific loci in response to biotic stress leads to an increased mutation rate at these loci due to the known higher frequency of G:methylated(C)→A:T transitions [6]. Beyond this, it was shown that biotic stress may trigger more global genetic changes that arise from the activation of transposable elements [7].

Plant genomes have exceptionally high levels of transposable element content and diversity [8]. A major group of mobile elements that may comprise up to 90% of plant genomes are LTR retrotransposons (LTR-RTEs) [9]. These elements replicate in genomes by transposition via RNA intermediates that are reversely transcribed by reverse transcriptase (RT) into cDNA and integrated into the genome by integrase complex [9]. While typically tens of thousands of LTR-RTEs are present in a genome, only some of them retain their transposition activity, shaping the mobilome of a plant cell [10]. Full-length LTR-RTEs are usually transcribed by RNA polymerase II (Pol II). Their transcription is suppressed by different epigenetic and small RNA-based mechanisms. The full-length and transpositionally active LTR-RTEs are often silenced by two RNA-directed DNA methylation (RdDM) pathways including PolIV/PolV (canonical) and RDR6/PolII/PolV (non-canonical) RdDMs. Importantly, a non-canonical RDR6-RdDM partially depends on post-transcriptional gene silencing (PTGS) components including RDR6, AGO1 and DCL1/2/4 enzymes [11]. The PTGS is also a crucial viral response mechanism in plants. Indeed, the vast majority of plant viruses possess RNA genomes, replicate exclusively in the cytoplasm, and are suppressed primarily by RNA interference (RNAi)—a specialized form of PTGS [12,13,14]. Thus, TE silencing and viral response mechanisms are significantly overlapped [14,15].

To counteract host defense mechanisms and enable efficient systemic spread, many viruses encode viral suppressors of RNA silencing (VSRs) that employ diverse strategies to interfere with host RNAi pathways. Among the most extensively studied VSRs is P19 derived from Tomato bushy stunt virus, which functions through siRNA sequestration. Additionally, P19 interacts with plasmodesmal-associated receptor-like kinases BAM1 and BAM2, thereby blocking the intercellular transport of siRNA and preventing the systemic spread of the silencing signal through the vascular system [16]. Through these mechanisms, P19 suppresses RNAi both locally, via siRNA sequestration, and systemically, by disrupting the intercellular movement of silencing components.

The current list of retrotransposons actively mobilized or transcriptionally induced in response to viral infection is quite limited. For instance, *Tnt1* LTR-RTEs with pathogen-inducible properties are triggered by a wide range of signals and stresses, including cucumber mosaic virus (CMV) [17]. In tomato, activation of *Retrolyc1* and *Kielia* has been observed during infection with Groundnut bud necrosis virus (GBNV) and Tomato leaf curl New Delhi virus (ToLCNDV). The expression of these retroelements correlates with the activation of genes related to programmed cell death and oxidative stress (for GBNV), as well as with the activation of phytohormonal pathways, including Auxin Responsive Factors and NAC1 (for ToLCNDV) [18]. In *Arabidopsis*, dynamic transcriptional activation of centromeric and pericentromeric *Ty1*/*Copia* and *Ty3*/*Gypsy* LTR-RTEs has been observed as a consequence of epigenetic control deregulation during Turnip mosaic virus (TuMV) infection [19]. Similar observations of transcriptional activation of LTR-RTEs (including *ATCOPIA4*, *AtGP1*, and *ATHILA2*) were noticed during Tobacco Rattle Virus (TRV) infection [20]. Thus, all of the above studies (excluding *Tnt1*) only demonstrate the transcriptional activation of retrotransposons in response to a virus. However, it remains unclear whether virus-induced transcription of LTR-RTEs leads to their transpositions generating novel insertions and structural variants.

In this study, we examined LTR-RTE activation in *Arabidopsis thaliana*, *Brassica napus* (rapeseed), and *Nicotiana benthamiana* infected with TRV, Potato Virus X (PVX), and Tobacco Ringspot Virus (TRSV). Using nanopore cDNA sequencing, we found that only PVX infection triggers transcription bursts of diverse sets of LTR-RTEs in *N. benthamiana*. To check the role of VSRs in this process, we performed a nanopore Mobilome-seq of *N. benthamiana* plants infected with a TRV virus encoding VSR P19. We identified a *Ty3*/*Gypsy Galadriel* LTR-RTE element that generates extrachromosomal circular DNA in these *N. benthamiana* plants. Altogether, the obtained results provide evidence that some viruses can trigger transcriptome as well as genome changes in the infected plants and suggest specific virus–host interaction mechanisms leading to genetic changes in response to biotic stress.

## 2. Results

### 2.1. Experimental Design and Virus Detection in Three Plant Species

To identify LTR-RTE activity in plants in response to viral infection, we used three positive-sense RNA plant viruses (Tobacco Rattle Virus (TRV), Potato Virus X (PVX), and Tobacco Ringspot Virus (TRSV) and three plant species (*Arabidopsis thaliana*, *Brassica napus* (rapeseed), and *Nicotiana benthamiana*). To initiate viral infection, we used cDNA clones of each virus inserted into binary vectors under control of the CaMV 35S promoter (Figure 1A). In addition, we used the TRSV vector encoding the GFP protein to visualize the systemic spread of the virus in different parts of the plant. *Agrobacterium* carrying the viral and VSRs binary vectors was delivered into leaves of the plants using the agroinfiltration method (Figure 1B). To distinguish the effects of agroinfiltration and VSR expression, plants infiltrated with *A. tumefaciens* carrying VSR binary vectors served as controls (Agro). Additionally, an infiltration buffer was used as a negative control (Buffer).

At 14 days post-infiltration (dpi), TRV and PVX triggered strong phenotypic responses in *N. benthamiana*, including reduced growth, leaf curling (TRV), and systemic chlorosis (PVX) (Figure 1C and Appendix A). Similar virus-specific symptoms were observed in *A. thaliana* at 14 dpi, with TRV inducing leaf curling and PVX causing chlorosis (Appendix A). In *B. napus*, plants were assessed at 7 dpi, at which point both TRV and PVX caused noticeable growth retardation (Appendix A). TRSV infection did not produce visible symptoms in any species. However, GFP fluorescence was detected in systemic leaves of *N. benthamiana* starting from 3 dpi, confirming viral movement (Figure 1C). No GFP signal was observed in *A. thaliana* or *B. napus*, and no symptoms were seen in Buffer or Agro control plants.

Systemic viral spread was checked in upper, non-infiltrated leaves by semi-quantitative RT-PCR analysis using two independent biological replicates (primer sequences provided in Appendix A). In *A. thaliana*, accumulation of TRV and PVX in systemic leaves was detected in all tested plants, while we observed no systemic TRSV infection. For *B. napus*, weak PCR products were obtained for TRV, PVX, and TRSV, indicating a low viral load. In *N. benthamiana*, systemic infection was confirmed for all three viruses (Figure 1D).

Thus, TRV and PVX established reliable infections across all plant hosts, whereas TRSV systemic infection was limited to *N. benthamiana*.

### 2.2. Transcriptome Response to Viral Infection

To assess the transcriptomic response to viral infection in the three plant species, we performed nanopore cDNA sequencing. This approach enables the detection of transcription from individual LTR-RTEs, as previously demonstrated in several plants, including *A. thaliana* and sunflower [21,22]. In total, we obtained between 1,561,763 and 4,110,106 high-quality nanopore reads per virus–host combination across two biological replicates. Using these data, we conducted a genome-wide transcriptome analysis of virus-infected plants relative to the Agro control (Appendix A). The analysis revealed that PVX triggered the most pronounced transcriptomic alterations, with 276 and 126 differentially expressed genes (DEGs) identified in *A. thaliana* and *N. benthamiana*, respectively (Figure 2). In contrast, no DEGs were detected in rapeseed plants infected with PVX.

TRV infection induced moderate transcriptomic changes across all three plant species, with *N. benthamiana* showing the lowest number of DEGs. Surprisingly, TRSV infection resulted in few or no DEGs in *A. thaliana*, *B. napus*, and *N. benthamiana*. Analysis of the identified DEGs revealed several shared expression patterns. For instance, *A. thaliana* and *N. benthamiana* plants infected with TRV both exhibited upregulation of glutathione S-transferase genes, which are known to respond to a wide range of biotic and abiotic stresses [23] (Appendix A). In contrast, these genes were downregulated in TRV-infected rapeseed plants. Both TRV and PVX infections induced significant bidirectional changes in the expression of several auxin-responsive genes.

In *N. benthamiana* infected with PVX, we also observed strong upregulation of numerous heat shock protein (HSP) genes, PLANT CADMIUM RESISTANCE 8, and the pathogen-related chitinase *Niben101Scf02171G00007.1* (Appendix A). Notably, unlike in *A. thaliana*, PVX infection in *N. benthamiana* led to the upregulation of cell wall biogenesis genes such as xyloglucan endotransglucosylase/hydrolases and cellulose synthase-like C5, suggesting species-specific differences in the transcriptional response to viral infection.

Thus, the transcriptome analysis demonstrated that viral infection modulates the expression of protein-coding genes in the studied plant species. Moreover, several expression changes such as the upregulation of glutathione S-transferases, heavy metal tolerance genes, and HSPs, along with the downregulation of jasmonate and ethylene biosynthesis and response pathways were observed in two or more species. The genes showing altered expression in response to viral infection have previously been reported as stress-responsive [24,25,26,27]. Collectively, these findings indicate that virus-responsive pathways are activated in the analyzed plants, providing a foundation for exploring whether such responses also trigger changes in LTR-RTE expression.

### 2.3. Numerous LTR Retrotransposons Are Transcribed Under Virus Stress

To determine whether viral infection promotes LTR-RTE transcriptional activation, we performed differential expression analysis across infected plants using the cDNA nanopore data and LTR-RTE annotations. Surprisingly, despite the detected viral load and transcriptomic changes, we found no LTR-RTEs with elevated expression in *A. thaliana* and *B. napus* plants infected by any of the viruses used. Similarly, no LTR-RTEs were upregulated in *N. benthamiana* infected by TRSV, despite its confirmed presence in systemic leaves and validated differential gene expression.

In contrast, differential expression analysis of LTR-RTEs of *N. benthamiana* plants infected by TRV and PVX identified 5 and 54 upregulated LTR-RTEs, respectively (Appendix A). Among these, 49 LTR-RTEs were activated exclusively in response to PVX infection, while 5 LTR-RTEs were detected in both TRV- and PVX-infected plants (Figure 3A). Phylogenetic classification of the activated *N. benthamiana* LTR-RTEs revealed members from three clades: *Galadriel* (*Ty3*/*Gypsy* superfamily), *Ivana* (*Ty1*/*Copia* superfamily), and *Alesia* (*Ty1*/*Copia* superfamily), indicating that viral-mediated derepression affects multiple retrotransposon clades (Figure 3B). Active LTR-RTEs in the *N. benthamiana* genome included both truncated (non-autonomous) and full-length (autonomous) elements. All three *Alesia* elements encoded only GAG, protease, and integrase, whereas approximately half of the *Ivana* (9 of 18) and *Galadriel* (17 of 33) elements were autonomous (Appendix A). Despite this structural diversity, full-length genomic RNA formation was detected for each element in at least one replicate of the PVX-infected *N. benthamiana* samples (Appendix A).

These results show that LTR-RTEs in different plant species display distinct expression patterns in response to various viral infections. Notably, the *N. benthamiana* genome contains a few LTR-RTE lineages whose expression is specifically upregulated following PVX infection.

### 2.4. Putative Regulatory Elements Within LTRs of Virus-Induced LTR-RTEs

We next analyzed the LTR sequences of actively transcribed LTR-RTEs to identify potential transcription factor binding sites. LTR sequences were extracted from all transcribed elements, except for one *Ivana* element that lacked identifiable LTRs. Representative LTRs from each clade were aligned, with *Galadriel* and *Ivana* elements further subdivided into two subgroups due to sequence divergence. Consensus sequences were generated for each of the five resulting alignments (Appendix A), and candidate transcription factor binding motifs were identified using a significance cutoff of *p* < 1 × 10^−4^ (Appendix A). We found a notable enrichment of *N. benthamiana* heat stress factor (HSF) recognition motifs in all active LTR-RTEs, including 4–10 HSF motifs in *Galadriel* elements, 8 in *Alesia* elements, and 8–20 in *Ivana* elements (Figure 4).

Motif analysis suggests that the observed transcriptional upregulation of LTR-RTEs in response to viral infection may result from the recognition of LTR sequences by HSF transcription factors, which are known to be activated by both abiotic and biotic stresses [28].

### 2.5. LTR-RTEs of N. benthamiana Produce Extrachromosomal Circular DNA in Response to a Viral Suppressor of RNA Silencing and Heat Stress

It is well-established that LTR-RTE activity in plants is suppressed by various epigenetic mechanisms and small RNA pathways [11,15]. Conversely, many plant viruses encode VSRs that inhibit small RNA pathways through different mechanisms [16]. Notably, PVX differs from TRV and TRSV in both the strength and function of its encoded VSR, P25. Therefore, we propose that the transcriptional upregulation of LTR-RTEs in response to PVX infection may result from the action of VSR suppressing LTR-RTE silencing pathways. To test this hypothesis, we examined the well-characterized P19 VSR encoded by Tomato bushy stunt virus (TBSV). P19 suppresses RNA silencing by sequestering small RNAs and inhibiting the RNA interference pathway. Importantly, the mode of action of P19 is highly similar to that of PVX P25, which also compromises RNA interference by promoting proteasomal degradation of AGO1/2/3/4 proteins.

The P19 sequence was delivered into the systemic leaves of *N. benthamiana* plants via agroinfiltration using TRV (TRV1 + TRV2:P19) into the first true leaves (Figure 5A). TRV was chosen as a vector because it did not cause significant transcriptional activation of *N. benthamiana* LTR-RTEs. Buffer-infiltrated plants and plants infiltrated by the TRV1 + TRV2 empty vector served as controls. To monitor LTR-RTE transposition activity, we employed nanopore sequencing (ONT Mobilome-seq [29]) of extrachromosomal circular DNAs (eccDNAs), which are typically generated during transposition events (Figure 5B). ONT Mobilome-seq of isolated and amplified eccDNAs from systemic leaves of *N. benthamiana* agroinfiltrated with TRV:P19 and control samples yielded over 150,000 reads per sample, with an N50 > 4 kb. Control samples infected by TRV lacking P19 did not show significant differences compared to the buffer-infiltrated control (Figure 5C, Appendix A), corroborating the poor LTR-RTE transcription changes in response to TRV infection. An eccDNA analysis revealed a significantly elevated eccDNA production from LTR-RTEs in the TRV:P19 sample (BH-adjusted *p* < 0.05). All P19-inducible LTR-RTEs belonged to the *Galadriel* clade and comprised four full-length and six truncated elements distributed across seven chromosomes (Figure 5C). Comparison of ONT cDNA and Mobilome-seq results revealed 4 elements presented in both datasets including three LTR-RTEs that were transcriptionally activated in response to PVX and one element (Chr13: 118,458,020–118,459,974) that was transcribed in plants infected by TRV as well as by PVX infection (Appendix A). Notably, among the active but previously untranscribed LTR-RTEs were tandem *Galadriel* elements located on chromosomes 6 and 19, which generated long eccDNAs of more than 8.5 kb and 6.8 kb, respectively.

The identified active *Galadriel* elements were also found to contain putative HSF-binding motifs within their LTRs (Section 2.4). Therefore, we examined whether elevated temperatures could also trigger activation of the P19-inducible *Galadriel* elements. We performed Mobilome-seq analysis on *N. benthamiana* plants exposed to heat stress (HS) and compared the levels of eccDNA generation with those in control plants not subjected to HS (Figure 5A,B). The results revealed that *Galadriel* elements produced eccDNAs in response to HS, although the level of eccDNA accumulation was lower than that observed in the TRV:P19 samples (Figure 5C). These findings demonstrate that *Galadriel* element activity can be modulated by both biotic and abiotic stresses.

To assess the relationship between eccDNA-producing *Galadriel* elements and other members of the clade, we performed a phylogenetic analysis based on 1129 reverse transcriptase (RT) sequences derived from *Galadriel* elements in our annotation (Appendix A). The resulting phylogenetic tree revealed five major clusters and several minor clusters (MC) (Figure 6A). Transcriptome and eccDNA sequencing data were then assigned to the phylogeny. This analysis indicated that the three eccDNA-producing elements, together with most transcribed elements (10 of 16 containing RT domain), were grouped within cluster 2. The remaining transcriptionally active elements were distributed across the minor clusters (number of elements *n* = 5), cluster 2 (*n* = 2), and cluster 4 (*n* = 1).

Further analysis investigated whether specific features of LTR-RTE sequences within individual clusters could account for the observed activity patterns. For all elements assigned to each cluster, distributions of LTR sequence identity (i.e., identity between LTRs of different elements within the cluster) and RT sequence identity to the RexDB reference sequence were calculated (Figure 6B, Appendix A). The median LTR sequence similarity for clusters 1 and 2 ranged from 94% to 97%. In contrast, most elements in clusters 3, 4, and 5 lacked identifiable LTR sequences despite retaining coding domains. The elements in clusters 1 and 2 exhibited the highest median RT sequence identity. However, none of the measured characteristics of cluster 2 elements exceeded those observed in the other active clusters.

Thus, we demonstrated a transposition burst of *Galadriel* LTR-RTEs in *N. benthamiana* induced by TRV:P19, highlighting the crucial role of viral suppressors of RNA silencing in activating specific transposon families and emphasizing the interplay between biotic and abiotic stresses and host genome mobilome dynamics.

## 3. Discussion

Viral infection induces profound alterations in the transcriptome, proteome, and metabolome of plant cells. However, the extent to which viral infections cause genomic changes, as well as the underlying molecular mechanisms remain poorly understood. In this study, we addressed this question by analyzing the activity of LTR-RTEs in three plant species following infection with three positive-strand RNA viruses (PVX, TRV, and TRSV). Although the LTR-RTE response was not universal across all species and conditions examined, we observed a significant transcriptional upregulation of LTR-RTEs in the *N. benthamiana* plants infected by PVX and TRV. Furthermore, using eccDNA nanopore sequencing, we provide direct evidence of the LTR-RTE retrotransposition process. Our findings indicate that LTR-RTEs may act as a molecular interface connecting viral infection, stress response pathways, and genome instability in plants. Comprehensive bioinformatic and experimental analyses suggest that virus-induced LTR-RTE activation results from the interplay between virus-triggered stress response mechanisms, specific transcription factor expression, and viral suppression of host small RNA pathways (Figure 7).

In this study, we initiated the viral infection of plants from two different plant families, Solanaceae (*N. benthamiana*) and Brassicaceae (*A. thaliana* and *B. napus*) using TRV, PVX, and TRSV viruses. We found that the transcriptomic responses to different viruses varied between the two plant families. In *N. benthamiana*, the transcriptome was primarily affected by PVX infection, whereas in *A. thaliana* and *B. napus,* major changes were associated with TRV infection. We also observed substantial intrafamily variation in the response to PVX: more than 200 DEGs were detected in *A. thaliana*, while no DEGs were identified in PVX-infected *B. napus* plants. Transcriptome analysis indicated the upregulation of stress-related genes, including glutathione S-transferases and those associated with cell wall development and remodeling in *A. thaliana* and *B. napus* upon infection. These genes are known to be involved in plant response to bacterial, fungal and viral infections [23,30]. The absence of transcriptomic changes upon PVX and TRSV infection correlated with weak or undetectable viral infection in *B. napus*. Surprisingly, the observed changes in the transcription of protein-coding genes in response to TRV (*A. thaliana* and *B. napus*) and even PVX (*A. thaliana*) infection did not correlate with the expression of LTR-RTEs in genomes of these species. This lack of mobilome response may be related to the specific characteristics of Brassicaceae transposons, which are activated under specific stresses. For example, in *A. thaliana* among intact LTR-RTEs, only the *ATCOPIA78*/*ONSEN* family is well-documented to be robustly activated by heat stress in wild type plants, and this appears to be conserved across Brassicaceae [31,32]. In our previous work, we applied different stresses in combination with epigenetic relaxation and also did not observe mobilome activation in the tested conditions except heat-stress during which *ONSEN* elements were transposed [33]. Thus, both *A. thaliana* and *B. napus* genomes may simply have no LTR-RTEs induced by the tested viruses. Another factor potentially explaining this lack of LTR-RTE transcriptional activation is low viral load. A direct link between transposon expression and viral load was previously demonstrated for *A. thaliana* infected with Turnip mosaic virus (TuMV) [19].

In contrast to *Arabidopsis* and *B. napus*, we observed a substantial, virus-specific increase in LTR-RTE expression in *N. benthamiana*. The set of elements induced by TRV almost completely overlapped with those activated by PVX. However, consistent with the more intensive transcriptional reprogramming and more severe phenotypic manifestations, PVX infection resulted in transcriptional activation of a substantially larger number of LTR-RTEs. We propose several cellular mechanisms that can explain this observation.

Firstly, LTR-RTE expression can be modulated by specific stress inducible transcriptional factors binding their LTRs, which serve as promoters for LTR-RTE transcription. Similar mechanisms have been shown for *ONSEN* transcription in heat-stressed plants. This activation of *ONSEN* occurs through the binding of HEAT STRESS FACTOR 2 (HSFA2) to the heat response element (HRE) in its LTRs [34]. Corroborating with this, all the identified *N. benthamiana* LTR-RTE elements contained HSF recognition motifs in their LTRs. Secondly, virus infection in plants induces extensive alterations in phytohormone levels, significantly modulating hormonal signaling pathways involved in plant defense and stress responses [35]. Hormone-responsive LTR-RTE transcription has been reported in many plants, such as *Arabidopsis*, tobacco, rice, barley, oat, sweet orange, strawberry, and tomato [36,37,38,39,40,41,42,43,44]. Thirdly, nearly all plant viruses encode VSR proteins that suppress PTGS and/or RdDM components as a defense strategy [45,46]. In addition to VSRs, some proteins essential for viral replication can interfere with transposon silencing machinery by other mechanisms. For example, the viral genome-linked protein (VPg) of multiple potyviruses can inhibit *SUPPRESSOR OF GENE SILENCING 3 (SGS3)*, a key PTGS component [47]. The role of *SGS3* in controlling LTR-RTEs has been demonstrated through its involvement in ribosome stalling [48]. Moreover, infection by Rice grassy stunt virus (RGSV) can lead to P3IP1-mediated degradation of OsNRPD1a, the core subunit of a key RdDM enzyme [49]. PVX used in this study encodes the viral suppressor P25 (also called TGBp1). P25 can lead to chronic endoplasmic reticulum (ER) stress and promote the accumulation of hydrogen peroxide (H_2_O_2_) in *N. benthamiana* cells [50,51]. H_2_O_2_, as a reactive oxygen species (ROS), is known to activate *ATCOPIA78*/*ONSEN* elements in *Arabidopsis*, likely due to heat-responsive motifs in their LTRs [52]. Another, more direct, effect of P25 is its action on PTGS components of a plant cell. P25 binds to AGO1/2/3/4 and leads to AGO1 depletion via proteasomal degradation of associated complexes [53]. Another viral suppressor P19 also reduces AGO1 levels through modulate accumulation of miR168 (regulates AGO1 mRNA), thereby interfering with AGO1-dependent silencing pathways [54]. AGO1 protein plays an important role not only in PTGS but also in transposon silencing pathways, as AGO1 together with RNA-dependent RNA polymerase 6 (RDR6) constitutes key components of the RDR6-RdDM pathway responsible for transposon suppression in plants [55,56]. Mutations in this pathway result in a substantial increase in the number of novel insertions of *EVD* LTR-RTE in the *A. thaliana ddm1* mutant [57]. Finally, viral infection can induce epigenetic changes in a genome that can result in epigenetic relaxation and transposon activation. In particular, TEs located proximally or within biotic stress response genes may be subject to virus-triggered demethylation [19,20]. This process is mediated by the deregulation of DNA methyltransferases (e.g., MET1) and DNA demethylases (e.g., ROS1), as demonstrated for TRV infection [58]. Additionally, viruses can alter the profile of 24-nucleotide small RNAs, including those targeting transposons and adjacent genes. This may shift the balance between the methylation and demethylation of transposons, which affects the function of genes adjacent to the transposons [58].

The results of our work may also be important for the application of virus-mediated technologies such as virus-induced gene editing (VIGE). Some VIGE protocols require the usage of tissue culture-based plant regeneration [59]. The plant tissue-culture itself can trigger transposition of LTR-retrotransposons [60,61,62]. Our results show that tissue-culture based VIGE can introduce not only targeted mutations but also novel LTR-RTE insertions, leading to genome and phenotype instability. One of the proposed approaches to limit such transposition is the use of RT inhibitors such as Tenofovir [63]. Despite these concerns, it is important to note that some VIGE applications have been successfully developed without tissue culture or with substantially reduced tissue culture requirements [64,65,66]. Nevertheless, taken into account that some viruses (e.g., TRV) can invade meristems, it can be assumed that particular cases of VIGE applications may face similar transposition challenges, even when tissue culture is minimized or eliminated.

## 4. Materials and Methods

### 4.1. Plant Growth Conditions

*Nicotiana benthamiana*, *Arabidopsis thaliana*, and *Brassica napus* plants were grown in a controlled growth chamber (Fujian Jiupo Biotechnology Co., Ltd., BPC600H, Fuzhou, China) at 22 °C under a 16 h light/8 h dark cycle.

### 4.2. Plants Agroinfiltration

Recombinant binary plasmids based on TRV1, TRV2, TRV2:P19, PVX, TRSV1, and TRSV2 were introduced into the *Agrobacterium tumefaciens* GV3101 strain by electroporation. Colonies were grown on solid LB with appropriate antibiotics at 28 °C for approximately 48 h, transferred to liquid LB, and cultured to OD_600_ 1.0–1.5. Cells were harvested (3500× *g*, 10 min), resuspended in infiltration buffer (10 mM MES, pH 5.6; 10 mM MgCl_2_; 100 µM acetosyringone), and incubated at room temperature for 2–3 h before infiltration. *Nicotiana benthamiana* (5–6-leaf stage; OD_600_ adjusted to 1.0), *Arabidopsis thaliana* (4–5-leaf stage; OD_600_ 1.5), and *Brassica napus* (4–5-leaf stage; OD_600_ 1.0) were infiltrated on the abaxial leaf side using needleless syringes. For TRV and TRSV, separate cultures of TRV1 and TRV2, or TRSV1 and TRSV2, were mixed 1:1 immediately before infiltration. PVX was infiltrated as a single construct. For all treatments described above, infiltration mixtures were supplemented with agrobacterial suspensions harboring p19 and P1/HC-Pro expression plasmids (each at 5% of the final suspension volume) to minimize RNA silencing. In a separate experiment, *N. benthamiana* (5–6-leaf stage; OD_600_ 1.0) was infiltrated with mixtures of separate cultures carrying TRV1 and TRV2 (1:1) or TRV1 and TRV2-P19 (1:1). Following infiltration, plants were maintained at 22 °C under a 16/8 h light/dark cycle. Fourteen days post-infiltration, apical systemic leaves were immediately collected, quickly frozen in liquid nitrogen, and stored at −80 °C for downstream analyses.

### 4.3. N. benthamiana Plants Heat Stress Treatment

Four-week-old *N. benthamiana* plants (6–7 leaf stage) were subjected to heat stress at 37 °C for 36 h. After completion of heat treatment, leaf tissues were immediately collected, quickly frozen in liquid nitrogen, and stored at −80 °C until analysis.

### 4.4. RNA Isolation, cDNA Synthesis, and PCR Validation

Approximately 100 mg of leaf tissue was homogenized in liquid nitrogen using a mortar and pestle. Total RNA was isolated using the ExtractRNA kit (Evrogen, Moscow, Russia) according to the manufacturer’s instructions. For cDNA synthesis, 1 µg of total RNA was reverse-transcribed using oligo(dT)_15_ primers and the MMLV RT kit (Evrogen, Moscow, Russia) following the supplier’s protocol. To validate systemic viral infection, PCR was performed on cDNA using virus-specific primers targeting genomic regions of TRV, PVX, and TRSV (Appendix A).

### 4.5. Double-Stranded cDNA Synthesis

For double-stranded cDNA (ds-cDNA) synthesis, 500 ng of total RNA was reverse-transcribed using oligo(dT)_15_ primers and the Mint kit (Evrogen, Moscow, Russia) according to the manufacturer’s instructions. The resulting ds-cDNA was purified using 1.8× volumes of Agencourt AMPure XP magnetic beads (Beckman Coulter, Brea, CA, USA) following the supplier’s protocol.

### 4.6. DNA Isolation

Genomic DNA was isolated using a modified CTAB protocol. Approximately 100 mg of fresh leaf tissue was ground in liquid nitrogen using a mortar and pestle. The frozen powder was mixed with 0.5 mL of preheated CTAB1 buffer (75 °C) supplemented with 6% β-mercaptoethanol and 0.5% polyvinylpyrrolidone (PVP). The lysate was transferred to a 1.5 mL microcentrifuge tube and incubated at 75 °C for 1 h. After cooling, an equal volume of chloroform was added, followed by centrifugation to separate the phases. The aqueous phase was transferred to a new 1.5 mL tube containing two volumes of CTAB2 buffer. The resulting DNA pellet was resuspended in 0.2 mL of 1 M NaCl, and DNA was precipitated by adding an equal volume of isopropanol and centrifuging. The pellet was washed with 70% ethanol and resuspended in nuclease-free water. RNA was removed by RNase treatment, followed by a second precipitation with isopropanol and an additional ethanol wash. The final DNA pellet was resuspended in nuclease-free water and stored for further analysis.

### 4.7. eccDNA Isolation and Amplification

eccDNA enrichment was performed according to a previously described protocol with minor modifications [67]. Briefly, 1 µg of total genomic DNA in a 50 µL reaction volume was treated with 1 µL (10 U) of Plasmid-Safe DNase (LGC Biosearch Technologies, E3101K, Beverly, MA, USA), 2 µL of 25 mM ATP, and 5 µL of 10× Plasmid-Safe reaction buffer. The reaction was incubated at 37 °C for 72 h with daily supplementation of 0.1 µL enzyme, 0.2 µL ATP, and 0.3 µL buffer. The enzyme was inactivated by heating at 72 °C for 30 min. Residual DNA was precipitated by overnight incubation with 1/10 volume of 3 M sodium acetate (pH 5.2) and 2.5 volumes of absolute ethanol, followed by centrifugation at 12,000× *g* for 30 min. Pellets were washed with ice-cold 70% ethanol and resuspended in 10 µL of nuclease-free water. eccDNA amplification was performed by random rolling circle amplification (RCA). The 20 µL RCA reaction contained 2 µL of phi29 DNA polymerase (Thermo Scientific, EP0091, Waltham, MA, USA), 2 µL of 10× phi29 buffer, 5 µL of 10 mM dNTPs, 1 µL of 500 µM exonuclease-resistant random primer (NpNpNpNpNpSNpSN, where *p* is a phosphodiester bond and pS is a phosphorothioate bond), and nuclease-free water. The reaction mixture was denatured at 95 °C for 5 min, then gradually cooled to 30 °C at a ramp rate of 1% using a thermocycler. RCA was carried out at 30 °C for 36 h, followed by enzyme inactivation at 65 °C for 10 min. For the debranching step, 500 ng of RCA product was treated with 1 µL of T7 Endonuclease I (New England Biolabs, M0302S, Ipswich, MA, USA) and 5 µL of 10× reaction buffer in a final volume of 50 µL. The reaction was incubated at 37 °C for 15 min and was stopped by adding chloroform, followed by extraction. The debranched product was precipitated with 1/10 volume of 3 M sodium acetate (pH 5.2) and 2.5 volumes of absolute ethanol, incubated at −80 °C for 30 min, and centrifuged at 12,000× *g* for 30 min. The final DNA pellet was washed with 70% ethanol and resuspended in nuclease-free water. For nanopore sequencing, 500 ng of eccDNA was used.

### 4.8. Library Preparation and Nanopore Sequencing

For nanopore sequencing of ds-cDNA and ONT Mobilome-seq, two libraries were prepared using Native Barcoding Kit 96 V14 (SQK-NBD114.96, Oxford Nanopore Technologies, Oxford, UK). Sequencing of ds-cDNA was performed on a Promethion P2 Solo equipped with a PromethION R10.4.1 flow cell.

ONT Mobilome-seq was performed using MinION equipped with an R10.4.1 flow cell. Basecalling was performed using Guppy 6.4.6 (Oxford Nanopore Technologies).

### 4.9. Genome Assemblies

For *A. thaliana*, TAIR10.1 (RefSeq accession: GCF_000001735.4) genome assembly and Araport11 genome annotation were downloaded from NCBI [68].

For *B. napus*, a Darmor-bzh v10 genome assembly and genome annotation were downloaded from the genoscope (http://www.genoscope.cns.fr/plants, accessed on 11 November 2025) [69].

For *N. benthamiana*, a NbT2T genome assembly was downloaded from the Zenodo database (https://zenodo.org/records/14010728, accessed on 11 November 2025) [70,71]. The genome annotation of N. benthamiana Niben101 (version 1.0.1) was obtained from the Sol Genomics Network (https://solgenomics.net/organism/1490/genome, accessed on 11 November 2025) [72]. The annotation was transferred to the NbT2T genome assembly using Liftoff v1.6.3 with the following parameters: mapping was performed with Minimap2, using the --end-bonus 5 and --eqx, with alignment options -N 50 -p 0.8 to increase sensitivity [73]. Gene models were lifted with a minimum alignment coverage of 0.8 (-a 0.8), sequence identity threshold of 0.9 (-s 0.9), and flanking region ratio of 0.1 (-flank 0.1). Only complete annotations were retained (-exclude_partial), while multi-copy genes were allowed (-copies), with a minimum sequence coverage of 0.95 (-sc 0.95) and an overlap threshold of 0.2 (-overlap 0.2). Gap penalties were set to -gap_open 2 and -gap_extend 1, and up to 2 mismatches were tolerated (-mismatch 2). Polishing of lifted models and coding sequence refinement were enabled (-polish -cds). Liftoff intermediates and unmapped features were recorded for quality control and validation.

### 4.10. LTR-Retrotransposons Annotation

For *A. thaliana*, the annotation of LTR-RTEs was downloaded from https://github.com/KaushikPanda1/AthalianaTETranscripts (accessed on 11 November 2025) [21].

*For B. napus* and *N. benthamiana*, the annotation of LTR-RTEs was performed using domain-based pipeline, with the detect_putative_ltr.R script from the DANTE_LTR toolkit v0.3.5.0 (https://github.com/kavonrtep/dante_ltr, accessed on 11 November 2025; with default parameters), followed by the domain-based DANTE tool v0.1.9 (https://github.com/kavonrtep/dante (accessed on 11 November 2025); with default parameters) [74], where the annotation was performed based on the Viridiplantae_v3.0 database (part of REXdb [75]).

### 4.11. Gene and LTR-Retrotransposons Expression Analysis

Ds-cDNA nanopore reads were aligned to the reference genome using Minimap2 (version 2.26) in splice mode (-ax splice) [76]. The resulting SAM files were converted to BAM format, sorted, and indexed using SAMtools (version 1.9). Primary alignments with a minimum mapping quality score of 30 were retained using the command samtools view -F 3840 -q 30 [77]. The filtered BAM files were used to quantify the gene expression levels with FeatureCounts (Subread package) [78]. Counting was performed based on the corresponding gene or LTR-RTE annotation data (see Section 4.9 for details on annotation processing) using the following parameters: -Q 30 -L --primary.

For differential gene expression analysis, read counts obtained from FeatureCounts were processed with the limma-voom algorithm via Galaxy (Version 3.58.1+galaxy0) [79]. Gene expression levels of virus-infected samples were compared to the control (Buffer). The Benjamini and Hochberg method of multiple test correction was applied for *p*-value adjustment (significance threshold = 0.05). Genes with an adjusted *p*-value < 0.05 were annotated using ShinyGO 0.85 [80].

For LTR-RTE expression analysis, read counts obtained from FeatureCounts were normalized to CPM. Expression levels were compared between the control (Buffer) and each virus-infected sample using a two-sided Fisher’s exact test (significance threshold = 0.01), implemented in the scipy.stats.fisher_exact function [81]. Multiple testing correction was applied using the Benjamini–Hochberg procedure with the statsmodels.stats.multitest.multipletests function [82]. Only LTR-RTEs with corrected *p*-values < 0.05 were retained for downstream analyses. LTR-RTEs with significant differences from the Buffer sample in at least one biological replicate were considered expressed.

### 4.12. Putative Transcription Factors Binding Motifs Prediction and Visualization

LTR sequences of actively transcribed transposable elements were extracted, except for one Ivana element lacking identifiable LTRs. Representative LTRs from each clade were aligned using mafft (Version 7.453) with parameters --thread 4 --threadtb 5 --threadit 0 --reorder --adjustdirection --maxiterate 0 --globalpair [83]. *Galadriel* and *Ivana* elements were further divided into two subgroups based on sequence divergence. The consensus sequence was derived using the dumb_consensus method with a 90% agreement threshold, assigning ‘N’ at ambiguous positions [84]. The fimo tool from the MEME package was used to search for motifs in the consensus sequence. The search was performed with the --thresh 1 × 10^−4^ parameters and the following fimo command parameters: --oc active_seq_scan --thresh 1 × 10^−4^. The transcription factor binding site set for *N. benthamiana* was downloaded from planttfdb.gao-lab.org (accessed on 11 November 2025) [85]. The coordinates of putative motifs were converted into quantities to construct a density plot (see Section 4.15).

### 4.13. Mobilome-Seq Analysis

The ONT Mobilome-seq reads were aligned to the TAIR10 reference genome using Minimap2 (version 2.26) with the parameter set -ax map-ont [76]. The resulting SAM files were converted to BAM format, sorted, and indexed using SAMtools (version 1.9). Primary alignments with a minimum mapping quality score of 30 were retained using the command samtools view -F 3840 -q 30 [77]. The sorted BAM files were inspected in a locally installed JBrowse2 genome browser [86].

To identify eccDNA-enriched genomic regions, the bedtools intersect command was applied to the LTR-retrotransposon coordinates and the corresponding BAM alignment files. The number of reads mapped to each LTR-RTE was compared between the control (Buffer) and virus-infected (TRV2 or TRV2:P19) samples using a two-sided Fisher’s exact test (significance threshold = 0.01), implemented in the scipy.stats.fisher_exact function [81]. Multiple testing correction was performed using the Benjamini–Hochberg procedure with the statsmodels.stats.multitest.multipletests function [82].

### 4.14. Phylogenetic Analysis

Amino acid sequences corresponding to reverse transcriptase (RT) domains were extracted from the DANTE annotation (see Section 4.10). Multiple sequence alignment was performed using MAFFT v7 with the parameters --reorder, --adjustdirection, --globalpair, and eight computational threads to ensure optimal alignment efficiency [83].

The resulting alignment was used to construct a maximum likelihood phylogenetic tree with IQ-TREE v2, employing automated model selection (-m TEST) and 1000 ultrafast bootstrap replicates (-bb 1000) combined with SH-aLRT support (-alrt 1000) [87].

### 4.15. Data Visualization

Euler and stacked bar chart visualization in Section 2.3 was performed with eulerr v.7.0.4 http://doi.org/10.32614/CRAN.package.eulerr (accessed on 11 November 2025) and ggplot2 v.4.0.0 R packages [88].

Density and sequence logo plot visualization in Section 2.4 was performed with matplotlib.pyplot and logomaker [89,90].

The phylogenetic tree was visualized and annotated using the online iTOL platform [91].

Dotplot visualization in Section 2.5 was performed with matplotlib.pyplot [89].

### 4.16. Plasmids

The TRSV1 and TRSV2-GFP plasmids were kindly provided by Chao Geng (Shandong Agricultural University, China) [92]. The pLX-PVX plasmid was kindly provided by José-Antonio Daròs (IBMCP (CSIC-Universitat Politècnica de València, Avenida de los Naranjos s/n 46022 Valencia, Spain) [93].

## Figures and Tables

**Figure 1 ijms-27-00286-f001:**
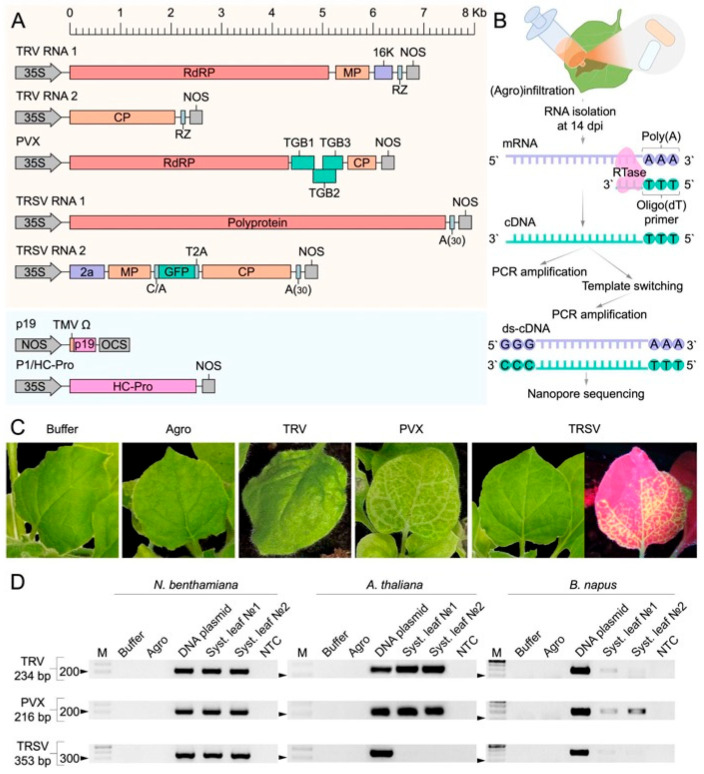
Experimental design, viral infection symptoms, and validation. (**A**) Linear representation of viral genomes and viral suppressors of silencing (VSRs) encoded in binary vectors. Abbreviations: 35S, CaMV 35S promoter; A(30), poly(A) sequence; RZ, hepatitis delta virus ribozyme; NOS, *nopaline* synthase terminator; 16K, RNA silencing suppressor from TRV; RdRP, RNA-dependent RNA polymerase; MP, movement protein; CP, coat protein; TGB, triple gene block proteins (PVX). (**B**) Schematic of the experimental workflow, including (agro)infiltration, RNA extraction at 14 days post-infection (dpi), cDNA synthesis, double-stranded cDNA (ds-cDNA) synthesis, and nanopore sequencing. (**C**) Phenotypic symptoms in systemic leaves of *N. benthamiana* infected with different viruses (TRV, PVX, and TRSV), compared to controls (buffer infiltration and *Agrobacterium tumefaciens* only). (**D**) Validation of virus presence by RT-PCR. Primer sequences are provided in Appendix A. Abbreviations: M, molecular weight marker, NTC, no template control.

**Figure 2 ijms-27-00286-f002:**
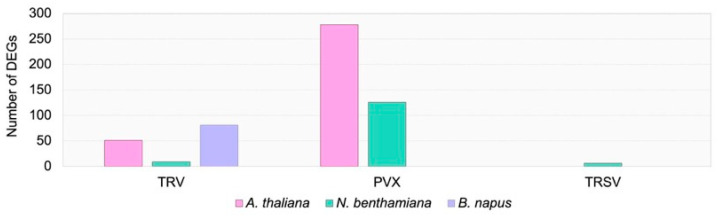
Number of DEGs in the three studied species compared to Agro controls. Bars are subdivided by color to indicate plant species: *A. thaliana* (pink), *N. benthamiana* (turquoise), *B. napus* (violet).

**Figure 3 ijms-27-00286-f003:**
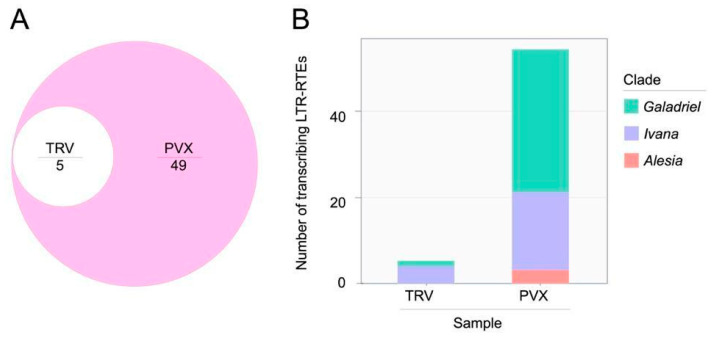
Analysis of *N. benthamiana* LTR-RTEs transcribing under viral infection. (**A**) Euler diagram showing the overlap of transcribed LTR-RTEs in *N. benthamiana* samples. (**B**) Stacked bar chart illustrating the number and clade classification of transcribed LTR-retrotransposons detected across two sample conditions: TRV and PVX. Bars are subdivided by color to indicate contributions from three phylogenetic clades: *Galadriel* (turquoise), *Ivana* (violet), and *Alesia* (red).

**Figure 4 ijms-27-00286-f004:**
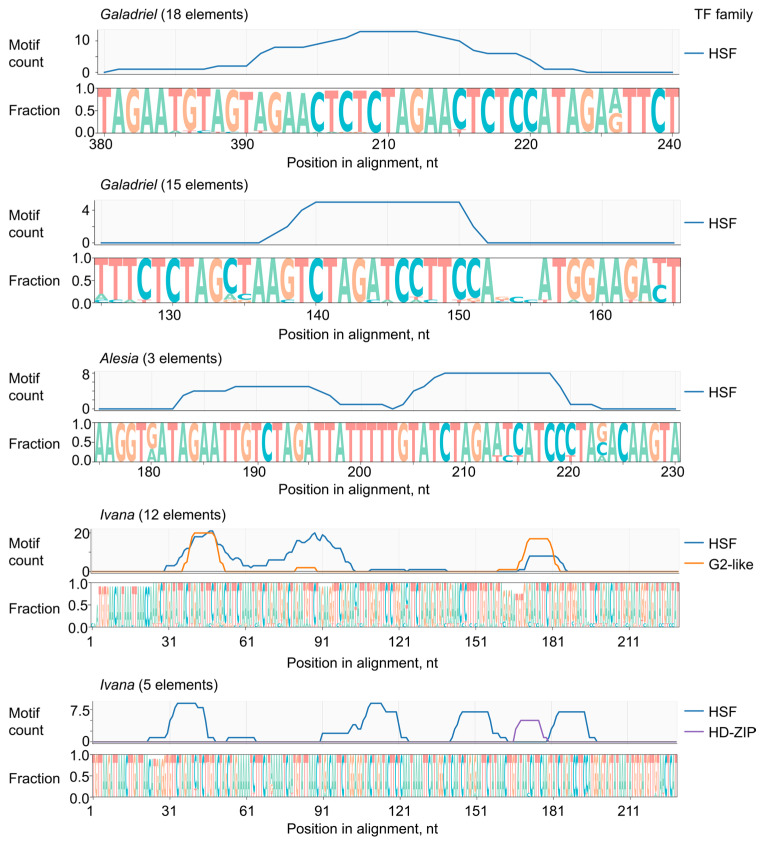
Logo and density plots of putative transcription factor binding motif positions aligned with the sequence logos of LTR regions from actively transcribed LTR-RTEs. Members of the *Galadriel* and *Ivana* clades are divided into two subgroups based on LTR sequence similarity. Only motifs with *p*-values < 1 × 10^−4^ are shown.

**Figure 5 ijms-27-00286-f005:**
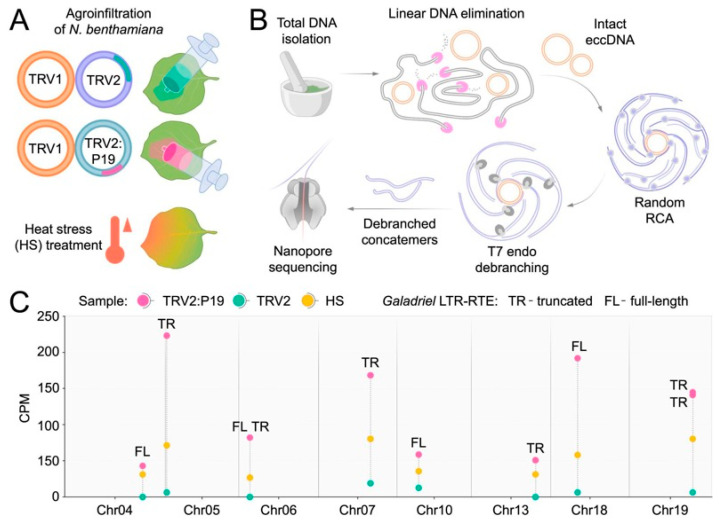
Experimental design and results of *N. benthamiana* ONT Mobilome-seq. (**A**) Experimental conditions used for mobilome activation in *N. benthamiana*. (**B**) Scheme of nanopore Mobilome-seq approach including DNA and eccDNA isolation with amplification and subsequent ONT sequencing. (**C**) Dot plot showing eccDNA counts of full-length (FL) and truncated (TR) *Galadriel* elements in TRV2:P19 (purple), TRV2 (turquoise), and heat stress (yellow). The *X*-axis indicates read abundance (counts per million, CPM); the *Y*-axis corresponds to chromosome position within the NbT2T assembly. Chromosomes without active LTR-RTs are not shown.

**Figure 6 ijms-27-00286-f006:**
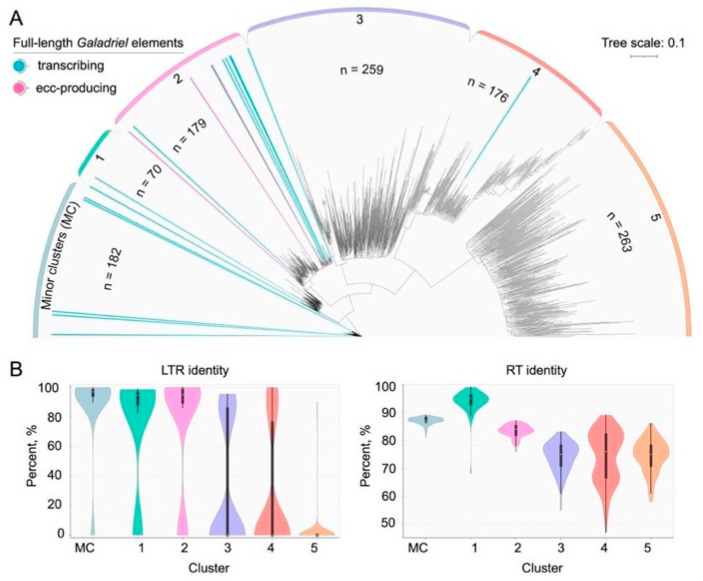
Phylogenetic analysis and characterization of *Galadriel* elements in NbT2T genome assembly. (**A**) Phylogenetic tree constructed based on reverse transcriptase (RT) sequences. Purple and turquoise lines indicate eccDNA-producing and transcribed LTR-RTs, respectively; the total number of elements in a cluster is denoted as “n”. (**B**) Violin plots characterizing representatives of individual clusters of the phylogenetic tree based on LTR and RT identity.

**Figure 7 ijms-27-00286-f007:**
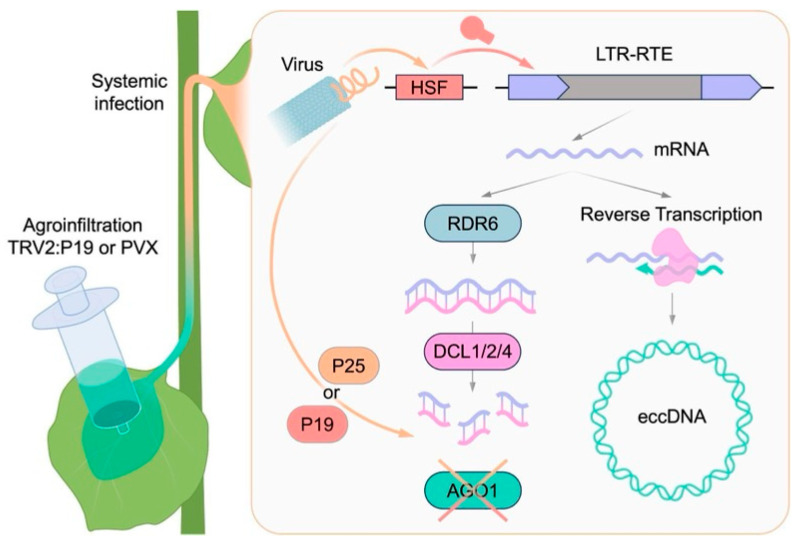
The proposed scheme for the observed virus-induced retrotransposition process in *N. benthamiana*.

## Data Availability

Nanopore data produced for this study are available in SRA (NCBI) under Bioproject Accession PRJNA1393863.

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
