# Peer review of "Positive-Strand RNA Viruses Induce LTR Retrotransposon Transcription and Extrachromosomal Circular DNA Generation in Plants"

_ijms, 2025, doi:10.3390/ijms27010286_

Round 1
Reviewer 1 Report
Comments and Suggestions for Authors
Positive-strand RNA [(+)ssRNA] viruses pose a serious threat to agricultural production and global food security. Although extensive studies have investigated the pathogenic mechanisms of (+)ssRNA viruses, the effects of viral infection on host genomes—particularly on retrotransposition—remain largely unexplored. Merkulov et al. analyzed transcriptional changes of long terminal repeat retrotransposons (LTR-RTEs) in plants infected by (+)ssRNA viruses, examined the roles of viral silencing suppressors (VSRs) in triggering the transcriptional burst of LTR-RTEs, and used Nanopore cDNA sequencing to assess the accumulation of extrachromosomal circular DNA (eccDNA) following infection. Their results provide initial evidence demonstrating that viral infection can promote not only retrotransposon transcription but also the retrotransposition process. These findings offer new insights into virus–host interactions and their co-evolution. Overall, the study raises several interesting points; however, the following issues need to be addressed before the manuscript can be considered for publication.
1. The authors state that it remains unclear whether virus-induced retrotransposon transcription leads to actual transposition events that generate novel insertions and structural variants. In this study, only eccDNA accumulation was analyzed. Although eccDNA is an intermediate of retrotransposition, as the authors themselves note, direct evidence is required to demonstrate that (+)ssRNA virus infection truly generates novel insertions or structural variants. The authors should provide genomic evidence showing new insertion events in host genomes following (+)ssRNA virus infection, rather than relying solely on eccDNA analysis.
2. The relationship between virus-induced genomic changes and the disease symptoms caused by (+)ssRNA virus infection warrants further discussion and analysis. Whether such genomic changes contribute to or correlate with symptom development remains unclear.
3. It is important to determine whether the genomic changes induced by (+)ssRNA viruses are heritable. The authors should assess whether progeny of infected plants retain virus-induced genomic alterations.
4. For Figures 3 and 6, it remains uncertain whether HSFs can indeed bind to the predicted motifs. Direct experimental evidence—such as ChIP-qPCR or EMSA—is required to validate the predicted HSF–DNA interactions.
5. Since the authors suggest that the retrotransposition burst is associated with VSR activity and RNA interference pathways, and given recent reports that TRV can be seed-transmitted (Liu et al., Cell Host & Microbe, 2024), the heritability of virus-induced genomic changes may be linked to seed transmission. In addition to dicots, the authors should also consider whether (+)ssRNA virus infection in monocots such as maize could induce retrotransposition bursts and potentially lead to more pronounced phenotypic consequences.
Author Response
Reviewer 1
We thank the reviewer for the comments. Please, find our point-by-point answers below. We believe these revisions have elevated the manuscript’s clarity and impact.
Q1. The authors state that it remains unclear whether virus-induced retrotransposon transcription leads to actual transposition events that generate novel insertions and structural variants. In this study, only eccDNA accumulation was analyzed. Although eccDNA is an intermediate of retrotransposition, as the authors themselves note, direct evidence is required to demonstrate that (+)ssRNA virus infection truly generates novel insertions or structural variants. The authors should provide genomic evidence showing new insertion events in host genomes following (+)ssRNA virus infection, rather than relying solely on eccDNA analysis.
A1. We thank the reviewer for this valuable comment. We agree that the term retrotransposition implies the occurrence of new genomic insertions, whereas our study did not provide direct evidence of such events. Detecting de novo insertions would require analysis of progeny or subsequent generations, which is beyond the scope of the current work and not feasible within this project. To avoid overstating our conclusions and to better reflect the actual focus of our study, we have revised the manuscript title to:
“Positive-strand RNA viruses induce LTR retrotransposon transcription and extrachromosomal circular DNA generation in plants”
We also modified the title of Section 2.5 in the Results to more accurately represent the study’s focus. Additionally, we carefully rephrased instances where retrotransposition was used, replacing it with a retrotransposition process to prevent misinterpretation of our findings.
Q2. The relationship between virus-induced genomic changes and the disease symptoms caused by (+)ssRNA virus infection warrants further discussion and analysis. Whether such genomic changes contribute to or correlate with symptom development remains unclear.
A2. Establishing a causal or correlative link between symptom development and eccDNA production is outside the scope of the current study. Viral symptom expression is associated with extensive host transcriptional reprogramming and stress responses that can influence plant phenotypes through multiple, potentially unrelated mechanisms. Dissecting the specific contribution of eccDNA generation would require targeted experimental approaches beyond the framework of this manuscript.
Q3. It is important to determine whether the genomic changes induced by (+)ssRNA viruses are heritable. The authors should assess whether progeny of infected plants retain virus-induced genomic alterations.
A3. We thank the reviewer for highlighting the importance of assessing the heritability of virus-induced genomic alterations. We agree that investigating whether such insertions are transmitted to the progeny would provide valuable insights into the long-term genomic consequences of (+)ssRNA virus infection. This complex and labor-intensive analysis lies beyond the scope of the present manuscript but represents a key direction for our future research. To improve clarity, we have also modified the title to emphasize that our study focuses on the retrotransposition process marked by eccDNA production rather than the fixation of novel genomic insertions.
Q4. For Figures 3 and 6, it remains uncertain whether HSFs can indeed bind to the predicted motifs. Direct experimental evidence—such as ChIP-qPCR or EMSA—is required to validate the predicted HSF–DNA interactions.
A4. We thank the reviewer for highlighting the importance of validating the predicted HSF–DNA interactions. While direct assays such as ChIP-qPCR or EMSA are beyond the scope of this study, we have strengthened our conclusions by incorporating new nanopore Mobilome-seq data from heat-stressed N.benthamiana plants (Section 2.5). These results provide strong indirect evidence that HSFs may contribute to the transcriptional activation of LTR-RTEs. Moreover, the same heat stress–activated LTR-RTEs are also induced by TRV:P19 infection, suggesting that Galadriel LTR-RTEs can be activated by both biotic and abiotic stresses (Figure 6 in updated version of manuscript).
Q5. Since the authors suggest that the retrotransposition burst is associated with VSR activity and RNA interference pathways, and given recent reports that TRV can be seed-transmitted (Liu et al., Cell Host & Microbe, 2024), the heritability of virus-induced genomic changes may be linked to seed transmission. In addition to dicots, the authors should also consider whether (+)ssRNA virus infection in monocots such as maize could induce retrotransposition bursts and potentially lead to more pronounced phenotypic consequences.
A5. We thank the reviewer for this thoughtful comment. Regarding the first point, the cited study (Liu et al., Cell Host & Microbe, 2024; https://doi.org/10.1016/j.chom.2024.08.009) reports seed transmission of cucumber mosaic virus (CMV, Cucumovirus family), but not tobacco rattle virus (TRV, Tobravirus genus). Although TRV can transiently invade meristematic tissues, its presence is typically limited to approximately 10 days post-inoculation (https://doi.org/10.1128/jvi.02438-07), and vertical transmission through seeds is not considered a canonical route for TRV spread. Moreover, as demonstrated by Liu et al. (2024), infection of stem cells within the shoot apical meristem by TRV does not result in heritable transmission to progeny.
Nevertheless, we agree that antiviral RNA interference (RNAi) mechanisms operating in vegetative tissues, the shoot apical meristem, and developing seeds may differ in their genetic requirements and sensitivity to viral suppressors of RNA silencing (VSRs). These differences could influence the potential heritability of virus-induced genomic changes.
With respect to the second point, extending such studies to monocot species, including maize, is indeed an exciting direction for future research aimed at understanding whether (+)ssRNA virus infections can trigger retrotransposition bursts with potential phenotypic consequences.
Reviewer 2 Report
Comments and Suggestions for Authors
This manuscript analyzed the activation of LTR-RTEs in Arabidopsis thaliana, Brassica napus, and Nicotiana benthamiana following infection with TRV, PVX, and TRSV. The result was found that PVX specifically triggered a strong transcriptional burst of diverse LTR-RTE families in N. benthamiana. Furthermore, through experiments involving TRV-P19, eccDNA derived from the Ty3/Gypsy Galadriel family was identified, confirming that viral infection can promote not only the transcription but also the retrotransposition process of LTR-RTEs. This manuscript provides direct evidence for retrotransposon-mediated genomic changes during plant–virus interactions. However, the following aspects of the paper require improvement:
- The text states that both TRV and PVX induced strong phenotypic responses in N. benthamiana, including reduced growth, leaf curling (TRV), and systemic chlorosis (PVX) in Figure 1C. However, the phenotypic differences between TRV-treated plants and controls in Figure 1C are not pronounced. It is recommended to provide clearer phenotypic images or quantitative data to strengthen this claim.
- The link between “transcriptome response” and "transposition burst" is insufficiently explained: The section "Transcriptome response to viral infection" primarily describes changes in host gene expression but does not adequately explain whythese changes directly contribute to the subsequent observed "transposition burst." It is recommended to add further analysis or discussion.
- The first part of the paper focuses on the systematic analysis of LTR-RTE transcriptional activation by three different viruses across various plant species, while the second part shifts to eccDNA detection mediated by TRV-P19. The logical connection between these two parts is weak. It is suggested that the Introduction or Discussion should further clarify why P19 was chosen as a representative VSR and compare its function with the VSRs of the three viruses used, thereby enhancing the overall coherence and logic of the study.
Author Response
We thank the reviewer for the comments. Please, find our point-by-point answers below. We believe these revisions have elevated the manuscript’s clarity and impact.
This manuscript analyzed the activation of LTR-RTEs in Arabidopsis thaliana, Brassica napus, and Nicotiana benthamiana following infection with TRV, PVX, and TRSV. The result was found that PVX specifically triggered a strong transcriptional burst of diverse LTR-RTE families in N. benthamiana. Furthermore, through experiments involving TRV-P19, eccDNA derived from the Ty3/Gypsy Galadriel family was identified, confirming that viral infection can promote not only the transcription but also the retrotransposition process of LTR-RTEs. This manuscript provides direct evidence for retrotransposon-mediated genomic changes during plant–virus interactions. However, the following aspects of the paper require improvement:
Q1. The text states that both TRV and PVX induced strong phenotypic responses in N. benthamiana, including reduced growth, leaf curling (TRV), and systemic chlorosis (PVX) in Figure 1C. However, the phenotypic differences between TRV-treated plants and controls in Figure 1C are not pronounced. It is recommended to provide clearer phenotypic images or quantitative data to strengthen this claim.
A1: We have reviewed this comment and corrected Figure 1C to provide more convincing, clear symptoms. We also included photos of the whole plants in the supplementary files.
Q2. The link between “transcriptome response” and "transposition burst" is insufficiently explained: The section "Transcriptome response to viral infection" primarily describes changes in host gene expression but does not adequately explain why these changes directly contribute to the subsequent observed "transposition burst." It is recommended to add further analysis or discussion.
A2. We thank the reviewer for this valuable comment. We agree that the link between the transcriptome response and the observed transposition burst required further clarification. In the revised manuscript, we have expanded this section to (1) provide additional evidence of virus-induced alterations in host transcriptomes and (2) emphasize that these transcriptional responses differ substantially between plant species and viral types. This distinction is important to highlight that the observed mobilome activation is not a direct reflection of overall transcriptomic changes, but likely results from specific stress- and virus-triggered regulatory mechanisms impacting transposon control.
Q3. The first part of the paper focuses on the systematic analysis of LTR-RTE transcriptional activation by three different viruses across various plant species, while the second part shifts to eccDNA detection mediated by TRV-P19. The logical connection between these two parts is weak. It is suggested that the Introduction or Discussion should further clarify why P19 was chosen as a representative VSR and compare its function with the VSRs of the three viruses used, thereby enhancing the overall coherence and logic of the study.
A3. We thank the reviewer for this insightful comment. We agree that the logical connection between the two parts of the study required clarification. In the revised manuscript, we have updated the Results section and added more detailed information about the P19 protein and other viral suppressors of RNA silencing (VSRs) in both the Introduction and Discussion sections to better explain the rationale for selecting P19 as a representative VSR and to strengthen the overall coherence of the study.
Reviewer 3 Report
Comments and Suggestions for Authors
The title of this manuscript sounds very interesting because it claims their work has shown that infection with several RNA viruses triggers many new insertions of retrotransposons into the host plant genome (i.e., a retrotransposition burst). However, the authors performed ONT Mobilome-seq of extrachromosomal circular DNAs (eccDNA) only from N. benthamiana plants infected with one virus (TRV:P19), showing an enhanced accumulation of eccDNAs mapped to 3 full-length and 6 truncated LTR retrotransposons (LTR-RTEs). Moreover, although eccDNA is often generated during retro-transposition, it remains unclear if eccDNA production is necessary and/or sufficient for retro-transposition. Thus, it is essential to sequence and analyze the plant genome to look for any new insertion event of LTR-RTEs, as shown for example in 2023 by Zhang et al (Nature Communications 14:5236). Because the authors did not demonstrate any new insertions of these retrotransposons in the genome of the infected plants by performing plant genome sequencing, the title and the main conclusion in the abstract “demonstrating that viral infection can promote not only retrotransposon transcription but also retrotransposition process” are over-interpretation and misleading.
The authors also investigated the transcriptional responses of several plant species to several positive-strand RNA viruses, including tobacco rattle virus (TRV) with or without expressing p19, a viral suppressor of RNA silencing from a different virus. Unfortunately, the results are poorly presented by using supplemental tables only. Moreover, the authors used agro-infiltration to initiate all viral infection together with agrobacterial suspensions harboring p19 and P1/HC-Pro expression plasmids. Since this inoculation procedure includes several artificial factors such as strong induction and suppression of transgene and antiviral silencing processes, all key results on the transcriptional responses must be verified by sap inoculation.
Other issues:
Table 1 was not found in the files submitted.
Fig. 1C: Is the leaf showing GFP from the infiltrated or non-infiltrated leaf?
Author Response
We thank the reviewer for the comments. Please, find our point-by-point answers below. We believe these revisions have elevated the manuscript’s clarity and impact.
Q1. The title of this manuscript sounds very interesting because it claims their work has shown that infection with several RNA viruses triggers many new insertions of retrotransposons into the host plant genome (i.e., a retrotransposition burst). However, the authors performed ONT Mobilome-seq of extrachromosomal circular DNAs (eccDNA) only from N. benthamiana plants infected with one virus (TRV:P19), showing an enhanced accumulation of eccDNAs mapped to 3 full-length and 6 truncated LTR retrotransposons (LTR-RTEs). Moreover, although eccDNA is often generated during retro-transposition, it remains unclear if eccDNA production is necessary and/or sufficient for retro-transposition. Thus, it is essential to sequence and analyze the plant genome to look for any new insertion event of LTR-RTEs, as shown for example in 2023 by Zhang et al (Nature Communications 14:5236). Because the authors did not demonstrate any new insertions of these retrotransposons in the genome of the infected plants by performing plant genome sequencing, the title and the main conclusion in the abstract “demonstrating that viral infection can promote not only retrotransposon transcription but also retrotransposition process” are over-interpretation and misleading.
Q1: We thank the reviewer for this important comment. We agree that demonstrating retrotransposition requires evidence of new genomic insertions, which was not assessed in our study. Therefore, we have revised the manuscript title and the corresponding text in the Results section (section 2.5) to avoid overstating our results and more accurately reflect what we studied. We have also replaced or clarified the use of the term "retrotransposition" in the manuscript to ensure that our findings are limited to retrotransposon transcription and eccDNA accumulation, rather than being interpreted as evidence of new insertion events.
Q2. The authors also investigated the transcriptional responses of several plant species to several positive-strand RNA viruses, including tobacco rattle virus (TRV) with or without expressing p19, a viral suppressor of RNA silencing from a different virus. Unfortunately, the results are poorly presented by using supplemental tables only.
A2. We appreciate the reviewer's feedback regarding the visualization of virus-induced transcriptome changes. We have now addressed this by providing a bar plot that displays the number of differentially expressed genes (DEGs) detected in each plant species (Figure 2 in updated version of manuscript).
Q3: Moreover, the authors used agro-infiltration to initiate all viral infection together with agrobacterial suspensions harboring p19 and P1/HC-Pro expression plasmids. Since this inoculation procedure includes several artificial factors such as strong induction and suppression of transgene and antiviral silencing processes, all key results on the transcriptional responses must be verified by sap inoculation.
A3. We appreciate the reviewer’s thoughtful comment. To control for the potential effects of agroinfiltration and VSR co-expression, we included Agrobacterium carrying binary vectors encoding VSRs as a control (“Agro samples”), as described in the Materials and Methods section. This control condition is now explicitly detailed in the Results section as well. To specifically assess the impact of viral infection on the transcriptome, all differential gene expression analyses were performed relative to the Agro control samples (see Results, section 2.2). Moreover, we analyzed only systemic leaves, which typically contain viruses but minimal residual Agrobacterium or VSR proteins. This design, maintaining the presence or absence of the viral genome as the sole variable between Agro and experimental samples, minimizes agroinfiltration-related confounding factors and ensures that the observed transcriptomic changes reflect the effects of viral infection itself.
Other issues:
Q3. Table 1 was not found in the files submitted.
A3: We have corrected the reference and added the corresponding data into Table S10 of the supplementary materials.
Q4. Fig. 1C: Is the leaf showing GFP from the infiltrated or non-infiltrated leaf?
A4. The leaf shown in Fig. 1C is a systemic (non-infiltrated) leaf. To clarify this, we have updated the figure caption accordingly. Additionally, we revised the text in the Results section (Section 2.1, paragraph 2) to explicitly state: "...GFP fluorescence was detected in systemic leaves of N. benthamiana starting from 3 dpi."
Round 2
Reviewer 2 Report
Comments and Suggestions for Authors
The resolution of images is relatively low and requires improvement.
Reviewer 3 Report
Comments and Suggestions for Authors
the authors have addressed the concerns I've raised.